

# Ecotoxicity of disinfectant benzalkonium chloride and its mixture with antineoplastic drug 5-fluorouracil towards alga *Pseudokirchneriella subcapitata*

Tina Elersek, Maja Ženko and Metka Filipič

Department of Genetic Toxicology and Cancer Biology, National Institute of Biology, Ljubljana, Slovenia

## ABSTRACT

**Background**. Benzalkonium chloride (BAC) is one of the most common ingredients of the disinfectants. It is commonly detected in surface and wastewaters where it can interact with the residues of pharmaceuticals that are also common wastewater pollutants. Among the latter, the residues of antineoplastic drugs are of particular concern as recent studies showed that they can induce adverse effect in aquatic organisms at environmentally relevant concentrations.

**Methods**. Ecotoxicity of BAC as an individual compound and in a binary mixture with an antineoplastic drug 5-fluorouracil (5-FU) was determined towards alga *Pseudokirchneriella subcapitata*, a representative of primary producers. The toxicity of the BAC+5-FU binary mixture was predicted by the two basic models: concentration addition (CA) and independent action (IA), and compared to the experimentally determined toxicity. Additionally combination index (CI) was calculated to determine the type of interaction.

**Results**. After 72 h exposure to BAC a concentration dependent growth inhibition of *P. subcapitata* was observed with an $EC_{50}$ 0.255 mg/L. Comparing the predicted no effect concentration to the measured concentrations in the surface waters indicate that BAC at current applications and occurrence in aquatic environment may affect algal populations. The measured toxicity of the mixture was higher from the predicted and calculated CI confirmed synergistic effect on the inhibition of algal growth, at least at $EC_{50}$ concentration. The observed synergism may have impact on the overall toxicity of wastewaters, whereas it is less likely for general environments because the concentrations of 5-FU are several orders of magnitude lower from its predicted no effect concentration.

**Discussion**. These results indicate that combined effects of mixtures of disinfectants and antineoplastic drugs should be considered in particular when dealing with environmental risk assessment as well as the management of municipal and hospital wastewaters.

Corresponding authors
Tina Elersek, tina.elersek@gmail.com
Metka Filipič, metka.filipic@nib.si

## INTRODUCTION

Quaternary ammonium compounds (QACs) are cationic surfactants that have been used for decades as disinfectants in numerous industrial, medical and domestic applications. These compounds are produced and used in large volumes; in EU the production of QACs per year exceeds 1,000 t (*European Chemicals Agency, 2014*) whereas their global annual consumption (together with amphoteric surfactants) has been estimated to be 1.2 million t in 2005 (*Uhl et al., 2005*). Due to such widespread use, QACs are common pollutants in wastewaters and receiving aquatic environment (*Zhang et al., 2015*). Structurally QACs have a nitrogen atom that is covalently bonded to four residues, making the nitrogen positively charged. One of the residues is typical an alkyl chain of length between about C5 and C18; two residues are methyl groups and the fourth residue is either a benzyl moiety or one more alkyl chain (*Wessels & Ingmer, 2013*). The toxicity of QACs is primarily based on their ability to disrupt membrane integrity via interaction with membrane lipids and/or transmembrane proteins, resulting in toxic effects for the exposed organisms (*Eleftheriadis et al., 2002*; *Tischer et al., 2012*). Ecotoxicological studies showed that QACs are highly toxic to different aquatic organisms (*Chen et al., 2014*; *Jing, Zhou & Zhuo, 2012*; *Kreuzinger et al., 2007*; *Liang, Neumann & Ritter, 2013*; *Nalecz-Jawecki & Grabinska-Sota E. Narkiewicz, 2003*; *Sanchez-Fortun et al., 2008*; *Zhu et al., 2010*). Moreover, certain QAC has been reported to induce genotoxic effects in mammalian and plant cells (*Ferk et al., 2007*) and in crustaceans (*Lavorgna et al., 2016*). Due to their ecotoxicological properties and ubiquitous presence in wastewaters and aquatic environment, QACs are of environmental concern. Furthermore, in the aquatic environment QACs do not occur as single compounds but in mixtures with other chemical contaminants (*Zhang et al., 2015*).

Pharmaceuticals for human and veterinary use are emerging contaminants that are regularly found in municipal and hospital wastewaters as well as in surface waters (*Hughes, Kay & Brown, 2012*). They enter the aquatic environment predominantly via the effluents from hospital and municipal wastewater treatment plants (*Ferrando-Climent, Rodriguez-Mozaz & Barcelo, 2014*), landfill leakages, and to a minor extent, in the discharge from the pharmaceutical industry. Among pharmaceuticals the residues antineoplastic drugs have been identified to be of particular environmental concern. They are regularly detected in hospital and municipal effluents, although their concentrations are relatively low in the ranges of ng/L to μg/L (*Kosjek & Heath, 2011*). However, due to their mechanisms of action most of these drugs are genotoxic, carcinogenic, and/or teratogenic and can cause adverse effects in aquatic organisms, especially upon chronic exposure (*Kuemmerer & Ruck, 2014*; *Toolaram, Kuemmerer & Schneider, 2014*). Several recent studies demonstrated that certain antineoplastic drugs cause adverse effects to different aquatic species even at concentrations relevant for aquatic environment (*Brezovšek, Eleršek & Filipič, 2014*; *Elersek et al., 2016*; *Kovacs et al., 2015*; *Kovacs et al., 2016*; *Misik et al., 2014*; *Parrella et al., 2014*; *Parrella et al., 2015*).

Although it can be expected that the residues of QACs and pharmaceuticals in hospital and municipal effluents occur together, to our best knowledge, ecotoxicity of the mixtures of the two groups of contaminants has not been studied so far. The aim of this study

was to assess the toxicity of representative QAC, benzalkonium chloride (BAC) alone and in a binary mixture with an antineoplastic drug 5-fluorouracil (5-FU) against green alga *Pseudokirchneriella subcapitata,* a representative of aquatic primary producers. BAC was selected as the representative of QACs because it is one of the most harmful QAC and among the most widely used antiseptic in hospitals and medical centres (*Ferk et al., 2007*). 5-FU, the representative of anticancer drugs, belongs to the most consumed antineoplastic drugs and has been recently also considered as a priority contaminant (*Booker et al., 2014*). The toxicity of the BAC+5-FU mixture was predicted using the concentration addition (CA) model (*Loewe & Muischnek, 1926*), and the independent action (IA) model (*Bliss, 1939*) and compared to the experimentally determined toxicity. The nature of the interaction was quantitatively evaluated by calculating combination index (CI) (*Chou & Martin, 2007*).

## MATERIALS & METHODS

### Tested chemicals

BAC (CAS 63449-41-2) and 5-FU (CAS 51-21-8) were obtained from Sigma-Aldrich (Seelze, Germany). Stock solutions of 5-FU (75 g/L) were prepared in dimethyl sulfoxide (DMSO; Merck, Darmstadt, Germany) and BAC (25 g/L) in OECD growth medium for algae, and both were stored at $-4\,°C$ in the dark not longer than 2 months. The experiments were conducted in compliance with internal safety standards for handling antineoplastic drugs (*Eitel, Scherrer & Kuemmerer, 2000*). All of the experimental residues were disposed of as hazardous waste.

### Test species

The test species, green alga *Pseudokirchneriella subcapitata* (SAG 61.81) (Fig. 1) was obtained from algae collection of the University of Goettingen (SAG). The alga was cultivated under the conditions defined in *OECD (2011)*.

### Toxicity studies—algal growth inhibition assay

The toxicities of BAC and the mixture of BAC+5-FU were determined according to *OECD (2011)*. BAC was tested in the range of concentrations from 0.008 to 1.950 mg/L (0.008; 0.023; 0.068; 0.203; 1.950) and from the concentration response curve the effect concentrations ($EC_x$) were calculated. The $EC_x$ values for 5-FU were obtained from our previous experiments in our laboratory (*Brezovšek, Eleršek & Filipič, 2014*). The mixture BAC+5-FU was tested at half $EC_x$ of each compound ($EC_5/2$, $EC_{10}/2$, $EC_{20}/2$, $EC_{50}/2$, $EC_{90}/2$) as described elsewhere (*Brezovšek, Eleršek & Filipič, 2014*; *Cleuvers, 2003*). For example, if BAC exhibit half of the effect at $EC_5$ (of BAC applied individually) and 5-FU exhibit half of the effect at $EC_5$ (of 5-FU applied individually) than together they exhibit effect at $EC_5$ (if they act individually). The cultures under exponential growth in OECD medium were exposed to BAC or BAC+5-FU mixture under continuous light (24 h per day; light intensity, 80–120 $\mu$mol photons/m$^2$ s), constant shaking, and the optimal temperature of $22 \pm 2\,°C$. The initial cell density was set at $5 \times 10^4$ algal cells/mL in 20 mL growth medium. The solvent DMSO was used at final concentration 0.02% v/v at all concentrations of the tested compounds and in control cultures. This concentration
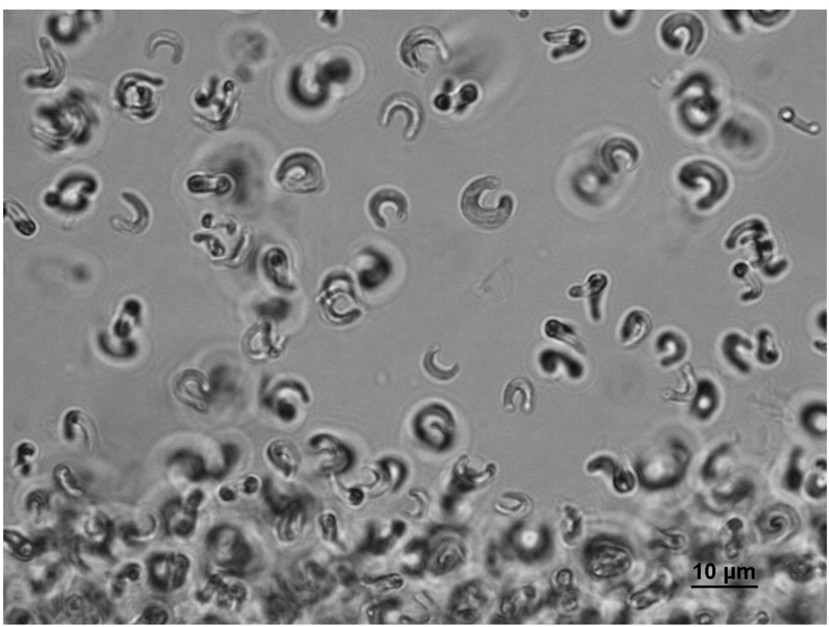

**Figure 1   Test organism *P. subcapitata* under light microcope.** Scale bar 10 μm.

has been previously demonstrated to have no effect on the growth of *P. subcapitata* (*Brezovšek, Eleršek & Filipič, 2014*).

During the exposure every 24 h the cell morphology was checked by light microscopy and growth was determined by removing 70 μL fractions from each test sample for cell counting by flow cytometry (*Elersek, 2012*). The observed endpoint was growth rate inhibition, expressed as the logarithmic increase in cell number (average specific growth rate) during the exposure period of 72 h. For each treatment three replicates were used and at least three valid independent experiments were used for statistical analyses.

## Data analysis, statistical evaluation and prediction of the binary mixture toxicity

The effect data were analysed with Prism 5 software (GraphPad, Inc., La Jolla, CA, USA) as measurement data from each individual flask pooled together (at least nine replicates) and plotted in an x,y graph, with x values transformed as log10. Outliers, recognised by Prism, were excluded from the analyses. Applied non-linear regression curve fit for effects was a sigmoid concentration-response fit with variable hill-slope (following Eqs. (1)–(3)) with the shared bottom value for all data set at control level. ECF is the concentration of the chemical that gives a response of F% between bottom and top (e.g., $EC_{10}$, $EC_{20}$, $EC_{50}$). Bottom and top are the plateaus in the units of the responses (% growth inhibition). The slope describes the steepness of the family of curves.

$$\log EC_F = \log EC_{50} + (1/\text{slope}) \times \log(F\%/(100 - F\%)) \tag{1}$$

$$F\% = (Y - \text{bottom})/(\text{top} - \text{bottom}) \times 100 \tag{2}$$

$$Y = \text{bottom} + (\text{top} - \text{bottom})/(1 + 10^{((\log EC50 - X) \times \text{slope})}) \tag{3}$$

Statistical significance ($p < 0.05$) of effect in comparison to the control was assessed by non-parametric ANOVA (Kruskal-Wallis test) with Dunn's post test at 95% confidence interval. The mixture values for top, hill-slope, $EC_5$, $EC_{10}$, $EC_{20}$, $EC_{50}$ and $EC_{90}$ were transferred to MS Excel software (Microsoft, USA) for predictive mathematical modelling, calculated according to two basic concepts (Eqs. (4) and (5)):

(i) Concentration addition—CA (*Altenburger & Greco, 2009*; originally from *Loewe & Muischnek, 1926*). CA was modelled by Eq. (4), where $X$ is the concentration of the mixture at which a specific effect occurs. $p_A$ is the fraction of chemical A in the mixture, $p_B$ the fraction of chemical B etc.; $x_A$ is the concentration level at which chemical A on its own exerts this specific effect. For a range of effect levels $x$ values were calculated, and a prediction curve was established.

$$X = (p_A/x_A + p_B/x_B + \cdots) \tag{4}$$

(ii) Independent action—IA (*Cleuvers, 2003*; originally from *Bliss, 1939*). IA was modelled by Eq. (5), where $E$ is the effect of the mixture at a specific concentration; $e_A$ is the effect of chemical A at that specific concentration and so-forth for chemical B etc. For a range of concentration points effects ($E$) were calculated, and a prediction curve was established.

$$E = 1 - ((1 - e_A)(1 - e_B)(\cdots)). \tag{5}$$

Combination index—CI (*Chou & Martin, 2007*; originally from *Chou & Talalay, 1984*) for multiple drug effect interactions was introduced for quantitative definition of synergism ($CI < 1$), additive effect ($CI = 1$), and antagonism ($CI > 1$) using computerized simulations. CI was modelled using CompuSyn software (ComboSyn, Inc., Paramus, NJ, USA) which is based on the CI concept with CI algorithms and median-effect equation (Eq. (6)). The ratio of the fraction affected ($fa$) vs. the fraction unaffected ($fu$) is equal to the dose ($D$) vs. the median-effect dose ($Dm$) to the $m$th power, where Dm signifies potency and m signifies the sigmodicity (shape) of the dose–effect curve.

$$(fa)/(fu) = [(D)/(Dm)]^m, \quad \text{where } fa + fu = 1. \tag{6}$$

Confidence interval of CI was assessed between 3 repeated experiments, each with three replicates. The 95% confidence interval at each level ($EC_5$, $EC_{10}$, $EC_{20}$, $EC_{50}$ and $EC_{90}$) was calculated by Monte Carlo technique described in *Bellen'kii & Schinazi (1994)* ($CI \pm 1.96 \times$ standard deviation).

## RESULTS

Exposure to BAC caused a concentration dependent growth inhibition of *P. subcapitata* (Fig. 2). Growth inhibition in this case can be calculated as >100% while treated culture exhibit toxicity through cell lyses and cell number goes below the starting number of cells. The effective concentrations (EC, Table 1) and the no effect concentration (NOEC, Table 1) calculated from the concentration response curve are in the range $\leq 0.1$ mg/L which according to classification of substances hazardous to aquatic environment (*UN, 2011*) classifies BAC as very toxic to aquatic environment.

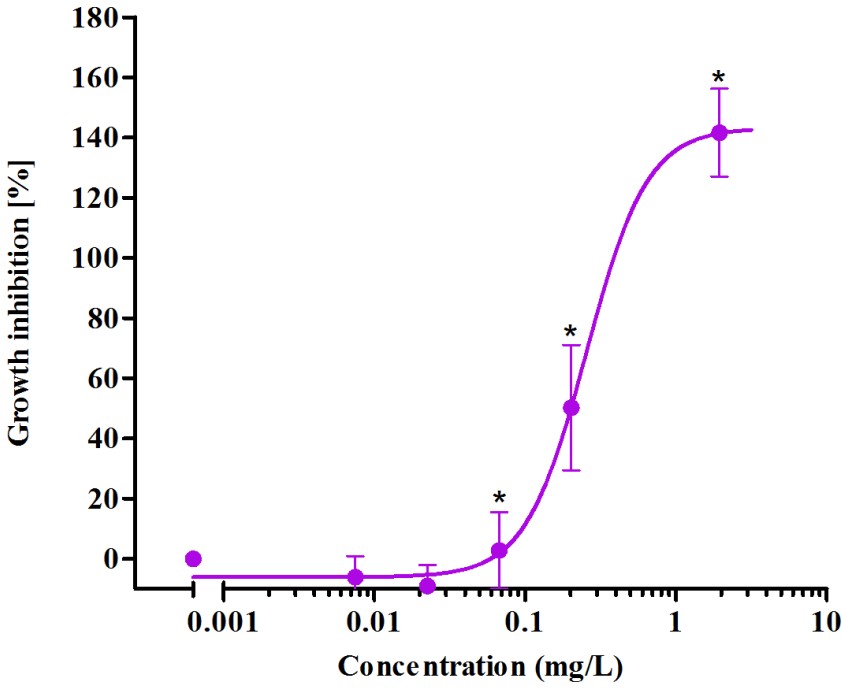

**Figure 2 Growth inhibition of alga *P. subcapitata* exposed to a disinfectant benzalkonium chloride [BAC].** Data represent means ± 95% confidence intervals, presented as log concentration response curves of growth rate inhibition after 72 h exposure. The significant differences compared to the control are marked with a symbol * ($p < 0.05$).

**Table 1 Ecotoxicological parameters of benzalkonium chloride (BAC) for growth inhibition of *P. subcapitata*.**

| Parameter | BAC (mg/L) | 95% confidence interval |
|---|---|---|
| EC$_5$ | 0.065 | (0.034–0.124) |
| EC$_{10}$ | 0.092 | (0.059–0.144) |
| EC$_{20}$ | 0.134 | (0.104–0.172) |
| EC$_{50}$ | 0.255 | (0.207–0.314) |
| EC$_{90}$ | 0.600 | (0.271–1.331) |
| NOEC | 0.023 | – |

**Notes.**
EC$_x$, effective concentration; NOEC, no observed effect concentration.

The growth inhibition of *P. subcapitata* by the binary mixture BAC+5FU was experimentally determined using half EC$_x$ of each individual compound (Table 2). The concentration–response curves generated from the experimental data were compared to the calculated predicted concentration response curves based on the CA model and the IA model (Fig. 3). Compared to individual compounds the binary mixture exerted higher growth inhibition at effect concentrations EC$_{20}$, EC$_{50}$ and EC$_{90}$, while at EC$_{90}$ the toxicity of the mixture was similar to BAC (Fig. 3). BAC and 5-FU are considered to act at different biological targets causing independent toxic effects therefore it is expected that IA model

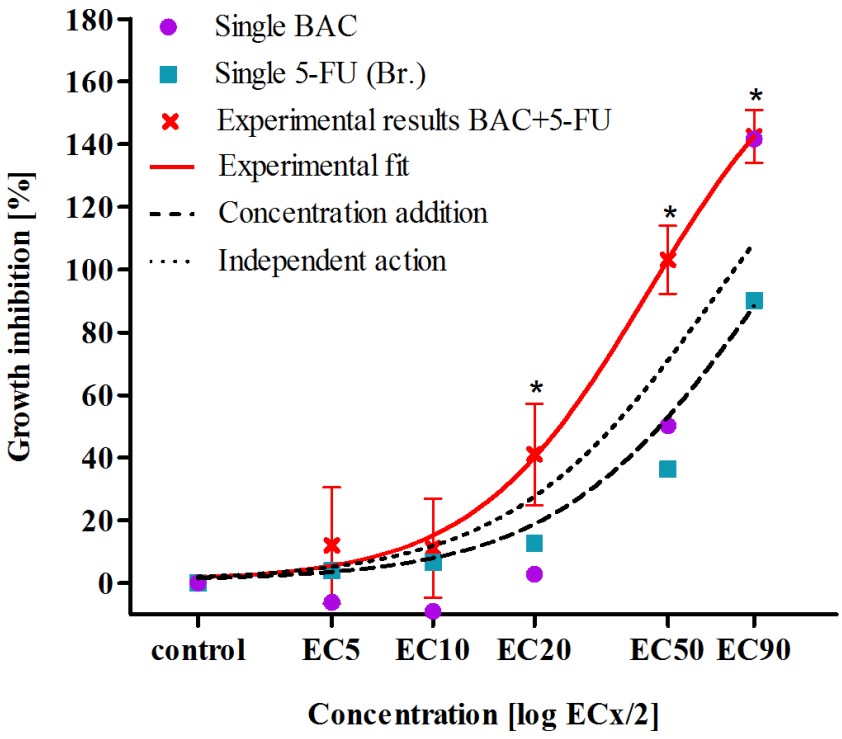

**Figure 3** Growth inhibition of alga *P. subcapitata* after exposure to the binary mixture of benzalkonium chloride [BAC] and 5-fluorouracil [5-FU]. Data represent means ± 95% confidence intervals, presented as log concentration response curves of growth rate inhibition after 72 h exposure, calculated from the experimental measurements and as predicted by the concentration addition model and the independent action model. The significant differences compared to the control are marked with a symbol * (*p* < 0.05). (Br., data from *Brezovšek, Eleršek & Filipič, 2014*).

**Table 2** Concentrations (mg/L) of benzalkonium chloride (BAC) and 5-fluorouracil (5-FU) in the BAC + 5-FU mixture according to their mixture effect concentrations ($EC_x$) for alga *P. subcapitata*. Individual 5-FU toxicity was investigated and described in a previous paper (*Brezovšek, Eleršek & Filipič, 2014*).

|                | BAC   | 5-FU  |
|----------------|-------|-------|
| $EC_5/2$       | 0.033 | 0.004 |
| $EC_{10}/2$    | 0.046 | 0.009 |
| $EC_{20}/2$    | 0.067 | 0.018 |
| $EC_{50}/2$    | 0.128 | 0.064 |
| $EC_{90}/2$    | 0.300 | 0.472 |

will better reflect the behaviour of the two compounds in the mixture. Nevertheless, the measured algal growth inhibition was at effective concentrations $\geq EC_{20}$ greater than that predicted by either IA or CA model (Fig. 3).

Combination index (CI) that characterize the deviation from the predicted outcome based on CA model showed synergistic effect BAC+5-FU (CI < 1) on algal growth inhibition which was most pronounced at $EC_{10}$ and $EC_{20}$ (Table 3). Different approaches (e.g., non-linear fitting, CA, IA, CI) exhibit some minor differences, but the synergistic

**Table 3 Combination index (CI) values with 95% confidence intervals for BAC+5-FU mixture for growth inhibition of *P. subcapitata* at different effect concentrations (EC$_x$).**

| | CI value | 95% confidence interval |
|---|---|---|
| EC$_5$/2 | 0.987 | (0.911–1.063) |
| EC$_{10}$/2 | 0.263 | (0.182–0.344) |
| EC$_{20}$/2 | 0.200 | (0.139–0.261) |
| EC$_{50}$/2 | 0.757 | (0.629–0.885) |
| EC$_{90}$/2 | 0.563 | (0.431–0.695) |

trend was approved by experimental results, which in our opinion gives more realistic information in comparison to any model or calculation.

## DISCUSSION

Benzalkonium chloride (BAC) is one of the most commonly used ingredients of hospital and domestic disinfectant that is regularly detected in hospital and municipal wastewaters together with the residues of pharmaceuticals. In this study we determined toxicity of BAC as an individual compound and in the mixture with an antineoplastic drug 5-FU against alga *P. subcapitata,* a representative of aquatic primary producers. The results confirmed that BAC is very toxic for *P. subcapitata* and demonstrated that in the binary mixture BAC+5-FU act synergistically on algal growth inhibition.

The effective concentrations (EC$_x$) and the no effect concentration (NOEC) of BAC (Table 2) are in the range <1 mg/L which, according to classification of substances hazardous to aquatic environment (*UN, 2011*) classifies BAC as very toxic to aquatic environment. For *P. subcapitata*, *Kreuzinger et al. (2007)* reported lower EC$_{50}$ value (0.04 mg/L), while similar EC$_{50}$ values have been reported for other algae *Chaetoceros gracilis* and *Isochrysis galbana* (*Pérez, Fernández & Beiras, 2009*). Similar or lower toxicity of BAC (EC$_{50}$ 0.3–3 mg/L) was observed in marine diatom species: (*Melosira nummuloides, Amphora coffeaeformis, Nitzschia incrustans, Navicula hansen, Cylindrotheca closterium, Achnanthes sp, Opephora sp., Navicula sp., Amphora sp.*) (*Beveridge et al., 1998*).

The residues of the antineoplastic drug 5-FU is has been regularly detected in hospital and municipal effluents (*Kosjek & Heath, 2011*). In our previous study we determined ecotoxicty of 5-FU towards *P. subcapitata* (*Brezovšek, Eleršek & Filipič, 2014*). The NOEC value was 0.01 mg/L, EC$_{10}$ 0.02 mg/L and EC$_{50}$ 0.13 mg/L, which is in the range of the values obtained with BAC.

The observed synergism may be explained by modes of action of the two compounds and their interactions in biological systems. BAC act mainly by disturbing the integrity and function of the cell membrane that leads to cell death (*Christen et al., 2017*). 5-FU inhibits thymidylate synthase and incorporation of its metabolites into DNA and RNA that block DNA synthesis and replication resulting in genetic damage and cell death (*Longley, Harkin & Johnston, 2003*). The synergistic effect of the mixture may result from alteration of the uptake of 5-FU via BAC mediated disruption of the cell membrane resulting in enhanced effect of 5-FU. Furthermore, BAC have been show to induce DNA and chromosome

damage (*Ferk et al., 2005*) indicating, that it can interact with genetic material although the mechanism its genotoxicity is not known. Thus, the synergistic action of BAC+5-FU mixture may also result from complementary interactions at the level of interaction with nucleic acids. However, further investigations concerning the molecular mechanisms responsible for this phenomenon are required.

We also aimed to elucidate if the presence of disinfectant BAC and its combination with the antineoplastic drug 5-FU in surface and wastewaters may lead to adverse environmental effects. We focused on growth inhibition towards alga, the representative of primary producers that play key role in the aquatic ecosystems. Algae are the base of aquatic food chains; consequently, death of algae will impact aquatic life on higher trophic levels (eg. secondary poisoning and food decrease). The concentrations of both compounds studied are quite low in surface waters; reported concentrations of BAC are mainly in the ranges around $\mu$g/L, (*Zhang et al., 2015*), while predicted environmental concentrations of 5-FU are in the range of ng/L (*Besse, Latour & Garric, 2012*). The predicted no effect concentrations (PNEC), which are derived from the $EC_{50}$ values divided by the assessment factor of 1000 (*EMA, 2006*), are for BAC 0.255 $\mu$g/L and for 5-FU 0.13 $\mu$g/L. Thus, it cannot be excluded that BAC may affect algal populations and consequently other aquatic organism. On the other hand, in municipal and hospital wastewaters BAC has been found at concentrations in the range of several hundred $\mu$g/L to mg/L (*Zhang et al., 2015*), and 5-FU at concentrations up to 124 $\mu$g/L (*Kosjek & Heath, 2011*), which implies potential risk to phytoplankton at local sites.

Regarding the effect of the binary mixture of BAC and 5-FU, this investigation demonstrated clear evidence that the combination lead to synergism as shown in Fig. 3 and Table 3. This indicates that the predictions of the effects of combinations of QACs and antineoplastic drugs may be underestimated when based on the ecotoxicological data for individual compounds. This is particularly important to take into account when assessing the toxicity of hospital wastewaters in which in addition to QACs also antineoplastic drugs may be present at concentrations higher than their PNEC values. However, it should be also noted that in this study we assessed the combined effect of the binary mixture over different effect concentrations that were for the two compounds similar, while in the hospital wastewaters the concentration ratios of the two compounds are very different. The concentrations of BAC are 1.000 to 10.000 fold higher from those of 5-FU. Therefore further studies are warranted to explore whether synergistic or potentiating effects are expressed also at the concentration ratios relevant for wastewater samples.

## CONCLUSIONS

The results of the present study confirmed that one of the most commonly used QACs, disinfectant BAC, is highly toxic for the alga *P. subcapitata*. Comparing the obtained PNEC values to the concentrations of BAC determined in surface and wastewaters indicate that at current uses and occurrence in the aquatic environment it can contribute to adverse environmental effects. The results of the experiments with the binary mixture of BAC and the antineoplastic drug 5-FU provided evidence that the two compounds act synergistically

on algal growth inhibition, which could not be predicted by CA and IA models. The observed synergism is particularly relevant for hospital and municipal wastewaters in terms of environmental risk assessment as well as wastewater management.

## ABBREVIATIONS

| | |
|---|---|
| BAC | benzalkonium chloride |
| 5-FU | 5-fluorouracil |
| CA | concentration addition |
| IA | independent action |
| CI | combination index |

## ACKNOWLEDGEMENTS

The authors would like to thank to Karmen Stanic and Kazimir Drašlar for their assistance with the experimental work.

### Funding

This study received funding from the Seventh Framework Programme FP7/2007-2013 under grant agreement No 265264 (CytoThreat) and by Slovenian Research Agency: the research core funding P1-0254. The funders had no role in study design, data collection and analysis, decision to publish, or preparation of the manuscript.

### Grant Disclosures

The following grant information was disclosed by the authors:
Seventh Framework Programme FP7/2007-2013: 265264.
Slovenian Research Agency: P1-0254.

### Competing Interests

The authors declare there are no competing interests.

### Author Contributions

- Tina Elersek conceived and designed the experiments, performed the experiments, analyzed the data, contributed reagents/materials/analysis tools, prepared figures and/or tables, authored or reviewed drafts of the paper, approved the final draft, analyzing data, nonlinear curve fiting and comparing to already published data.
- Maja Ženko conceived and designed the experiments, performed the experiments, analyzed the data, prepared figures and/or tables, authored or reviewed drafts of the paper, approved the final draft, analyzing data and comparing to already published data.
- Metka Filipič conceived and designed the experiments, contributed reagents/materials/analysis tools, authored or reviewed drafts of the paper, approved the final draft, interpreting the possible mechanisms of synergistic action.

## Data Availability

The raw data are provided in a Supplemental File.

## Supplemental Information

Supplemental information for this article can be found online at http://dx.doi.org/10.7717/peerj.4986#supplemental-information.

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
