# Peer review of "Ecotoxicity of disinfectant benzalkonium chloride and its mixture with antineoplastic drug 5-fluorouracil towards alga Pseudokirchneriella subcapitata"

_PeerJ, doi:10.7717/peerj.4986_

## Round 0.1 · original submission · Major Revisions

The Reviewers have made suggestions and comments which, I believe, will help to improve the manuscript. Please pay special attention to the comments on the statistical analysis, information about the number of replicates and variability of the data. Please make appropriate changes and submit your revised manuscript.

Reviewer 1 ·

Basic reporting

The article text is supported by the appropriate literature, however some citations need to be corrected:
(a) paper by Abel (1974) is not good citation for membrane-QAC interaction. I would recommend to change it. It may be e.g. Tischer, M., Pradel, G., Ohlsen, K., & Holzgrabe, U. (2012). Quaternary ammonium salts and their antimicrobial potential: targets or nonspecific interactions?. ChemMedChem, 7(1), 22-31 (or something similar);
(b) Kosjek and Heath (2011) is incorrectly cited in the text as “Kosjek et al.”;
(c) Kuemmerer and Ruck (2014) is incorrectly cited in the text as “Kuemmerer et al.”;
(d) Altenburger and Greco (2009) is incorrectly cited in the text as “Altenburger et al.”;
(e) Kreuzinger et al. (2007) is incorrectly cited in the text as “Kreuzineger et al.”.

Figures
(a) Figure 1 – right photo (electron microscope) was already published in Authors paper (2016) but it is not stated in the figure caption. In my opinion this figure is not essential for publication so it may be removed. Otherwise it should be described as reprinted from the other paper;
(b) Figure 2 – please, give the information how many replicates did you use to calculate mean values.

Tables. In Table 1 description it should be stated that 5-FU toxicity was investigated earlier and described in the previous Authors paper. There is a misspelling in the word “benzalconium”.

Experimental design

It should be clearly stated in the “Toxicity studies – algal growth inhibition assay” section, that 5-FU toxicity was investigated earlier and described in the previous Authors paper (it is only mentioned in Discussion and in Fig. 3).

Validity of the findings

No negative comments.

Additional comments

The problem presented in the manuscript is very interesting and environmentally important. The results could be considered as a valuable broadening and supplementing of the previous Authors’ works. The manuscript is well prepared and needs only minor improvement.

Reviewer 2 ·

Basic reporting

Article could use minor revisions for wording for professional English throughout.

Introduction (general): Introduction needs more citations for the points the authors make. QACs are common pollutants in wastewaters (Lines 53-54); pharmaceuticals are commonly found in municipal wastewaters (70-71); they enter the environment via effluents… (71-74).

Experimental design

Line 102: Did authors use a solvent control to control for possible effects of DMSO on algal growth?
Line 116-119, and Table 1 and 2: The concentrations of BAC used in this study need clarification. Based on the text, authors chose their concentrations for the mixture toxicity based on previous work by other authors. However, the concentrations listed as used for BAC are exactly half the relevant ECX values shown in Table 2. So were the concentrations chosen based on previous studies, or this study, or did the previous studies result in exactly the same point estimates as the current one?
Line 118: Why was 5-FU not retested on its own in the current study? Authors are relying on data from a 15 year old study (Cleuvers 2003) performed by a different laboratory?
Line 135: How many replicates were used for each treatment?

Validity of the findings

Line 225: It is unclear what the authors mean by "we observed heterogeneity". Please reword.
Lines 263-273: Too much info from the previous study. Not necessary.

Additional comments

Line 26: Calculated EC50 was 0.255, not 0.3. No reason to round up in the abstract.
Lines 52-53: Units of t versus kt. Change one of these to be consistent.
Line 81: Needs rewording, or another comma. "…even at low". Also, what is low here?
Line 88: Is 5-FU an antincancer drug or an antineoplastic?
Table 1: Typo in caption "benzalconium". Third column in the table is unnecessary
Table 2: If BAC concentrations used in Table 1 are based on ECX values from Table 2, it stands that the results in Table 2 should be discussed first, and thus the two tables should switch spots.
Table 3: Are there statistics associated with these CI values? Or confidence limits of any sort?
Figure 1: Photos are too dark. Scale bar not visible on left photo.

---

## Round 0.2 · accepted · Accept

Thank you for carefully revising your manuscript and addressing the reviewers' suggestions. It has been a pleasure working with you.

#